# Leveraging information between multiple population groups and traits improves fine-mapping resolution

Feng Zhou [1], Opeyemi Soremekun[2], Tinashe Chikowore [3,4,5,6], Segun Fatumo [2,7], Inês Barroso [8], Andrew P. Morris [9] & Jennifer L. Asimit [1]

Statistical fine-mapping helps to pinpoint likely causal variants underlying genetic association signals. Its resolution can be improved by (i) leveraging information between traits; and (ii) exploiting differences in linkage disequilibrium structure between diverse population groups. Using association summary statistics, MGflashfm jointly fine-maps signals from multiple traits and population groups; MGfm uses an analogous framework to analyse each trait separately. We also provide a practical approach to fine-mapping with out-of-sample reference panels. In simulation studies we show that MGflashfm and MGfm are well-calibrated and that the mean proportion of causal variants with PP > 0.80 is above 0.75 (MGflashfm) and 0.70 (MGfm). In our analysis of four lipids traits across five population groups, MGflashfm gives a median 99% credible set reduction of 10.5% over MGfm. MGflashfm and MGfm only require summary level data, making them very useful fine-mapping tools in consortia efforts where individual-level data cannot be shared.

Many genetic associations have been identified through genome-wide association studies (GWAS)[1], but little is known about the underlying mechanisms that drive such associations. Translation of these findings into new therapeutic targets or revealing new biological insights for diseases is aided by statistical fine-mapping. Fine-mapping identifies potential causal variants that underlie the genetic associations with the aim of reducing the number of genetic variants for follow-up in downstream functional validation experiments[2,3].

In genetic studies, we often group people based on genetic similarity, to avoid problems of structure in the data that could lead to false-positives. Exploiting differences in linkage disequilibrium (LD) across different genetically similar groups ascertained from diverse populations could lead to improvements in fine-mapping resolution[4–6].

We use the term "group" to refer to a genetically similar group of individuals, which may be from one or more cohorts/studies that contribute to a GWAS, either based on a single-study or meta-analysis. There are two main approaches to multi-group fine-mapping: (i) approaches that assume a single causal variant and account for heterogeneity in allelic effects between GWAS[7,8]; (ii) approaches that allow for multiple causal variants at a locus and use LD from each group to model joint effects of variants, either assuming homogeneous effects between GWAS[9] or allowing heterogeneous effects[10].

Approaches that assume a single causal variant have the advantage of not needing LD information to model joint variant effects and they include variants that are present in at least one of the groups. This can lead to an imbalance in sample sizes for each variant, such that

[1]MRC Biostatistics Unit, University of Cambridge, Cambridge, UK. [2]The African Computational Genomic (TACG) Research Group, MRC/UVRI and LSHTM, Entebbe, Uganda. [3]Sydney Brenner Institute for Molecular Bioscience, Faculty of Health Sciences, University of the Witwatersrand, Johannesburg, South Africa. [4]MRC/Wits Developmental Pathways for Health Research Unit, Department of Paediatrics, Faculty of Health Sciences, University of the Witwatersrand, Johannesburg, South Africa. [5]Channing Division of Network Medicine, Brigham and Women's Hospital, Boston, MA, USA. [6]Harvard Medical School, Boston, MA, USA. [7]Department of Non-Communicable Disease Epidemiology, London School of Hygiene and Tropical Medicine, London, UK. [8]Exeter Centre of Excellence for Diabetes Research (EXCEED), University of Exeter Medical School, Exeter, UK. [9]Centre for Genetics and Genomics Versus Arthritis, Centre for Musculoskeletal Research, University of Manchester, Manchester, UK. ✉e-mail: jennifer.asimit@mrc-bsu.cam.ac.uk

variants present in larger groups tend to be favoured, even though they may not be present in all groups. In contrast, current approaches that allow for multiple causal variants only consider variants that are present in all of the groups, but this could exclude an increasing number of variants when population diversity between groups is high and as the number of groups increases. For example, when GWAS are ascertained from diverse populations, some variants may be common or low-frequency in some groups, and monomorphic in others. In current multiple causal variant approaches, such variants would be removed from all groups, but this may potentially impact the multi-SNP model. That is, in an attempt to account for trait variability explained by the missing causal variant(s), several non-causal variants may be added into the multi-SNP model, giving spurious results.

Among the many reported genetic associations, examples of pleiotropy, where a gene affects several phenotypes, are widespread, possibly due to a shared variant affecting a pathway involved in multiple related traits and diseases[11]. When multiple traits have signals in the same region, colocalisation is often used to assess the evidence that two (or more) traits share the same causal variant (and sometimes identifies the variant). However, fine-mapping aims to identify causal variants, which may or may not be shared across traits, but it takes advantage of the correlation among traits to improve localisation. Colocalisation methods often make simplifying assumptions of at most one causal variant and uncorrelated traits. For example, mcoloc[12] does not consider correlations between traits because it requires that all traits are measured in distinct datasets of unrelated individuals. In addition, colocalisation methods are designed for only a single population group.

Multi-trait fine-mapping methods that leverage information between correlated traits could improve fine-mapping resolution, as biologically related traits often share causal variants[13]. Few multi-trait fine-mapping methods exist due to the computational challenge of many possible combinations of models (allowing multiple causal variants) between traits. One approach is to reduce the search space of joint causal variant models between traits by assuming that traits share all causal variants, allowing for different effect sizes, as in fastPAINTOR[14]. Alternatively, flashfm[13] jointly fine-maps multiple traits without the restriction of shared causal variants between traits; it shares information between traits by upweighting prior probabilities for joint trait models that have a shared causal variant. Flashfm gains computational efficiency in two ways: (i) partitioning the joint Bayes' factor into marginal components that depend only on the individual trait datasets, avoiding the need to store many joint model results; and (ii) reducing the search space of joint models by learning from single-trait fine-mapping results.

For fine-mapping methods that allow multiple causal variants, LD information is needed for each group. It is ideal to have access to the genotype data of the GWAS for each group as a source of LD, but this is often not possible. A practical alternative is to use a reference panel (RP) for each group, such that the RP closely matches the group. Each group-specific RP may either be a subset of the GWAS of the group, such as a cohort that contributed to the meta-analysis, or external, out-of-sample data that are from genetically similar groups. The 1000 Genomes data is an easily accessible source of genotype data, providing RPs that are classified as African, European, East Asian, South Asian, and Admixed American, based on their genetic similarities, with average sample size of 500[15], but its use with large GWAS may have issues[16].

In this work, we provide a practical strategy for using the 1000 Genomes data with GWAS summary statistics, advocating for a parsimonious model (a model with few causal variants) when using any of the available summary-level fine-mapping methods. Our dynamic algorithm adjusts the maximum number of causal variants, as learned by the data. Exploiting information from multiple traits and groups, we introduce a multi-group multi-trait fine-mapping approach, MGflashfm, and an analogous framework for multi-group fine-mapping of a single-trait (MGfm). Both approaches allow multiple causal variants and include variants that appear in at least one group, rather than using the intersection of variants across groups. In simulation studies, we compare MGflashfm and MGfm with current multi-group fine-mapping approaches that allow multiple causal variants, PAINTOR[9] and msCAVIAR[10], and also with a multi-trait fine-mapping method, mvSUSIE[17]. We fine-map genetic associations among four lipids traits in five different genetically similar groups from the Global Lipids Genetics Consortium (GLGC)[18] using MGflashfm and MGfm and compare their 99% credible sets in terms of size and variant posterior probabilities (PP); functional annotations of variants with PP > 0.90 are also provided.

## Results

### Multi-group fine-mapping conceptual framework

To enable summary statistics-based fine-mapping across multiple traits and multiple groups we developed MGflashfm, extending the framework of flashfm[13] multi-trait fine-mapping. For each trait, flashfm outputs the top SNP models and the model posterior probability (PP), adjusted for information from the other traits. This means flashfm results can be used comparably to those from single-trait fine-mapping, but with an expectation of greater precision.

We extend flashfm to multiple groups by taking advantage of the independence between the groups, which are assumed to have no sample overlap. The GWAS for each group may be based on a single study or a meta-analysis of multiple cohorts that are genetically similar. We developed MGflashfm to allow for up to six groups, therefore we recommend that studies having a similar LD pattern are meta-analysed prior to use. MGflashfm allows for variants that are not present in all groups, serving two purposes: (i) it retains causal variants that may not be observed in all groups because of low-frequency or being monomorphic, and so are group-specific causal variants; and (ii) it retains variants that are causal in multiple groups, but do not pass quality control in all groups.

To account for the group-specific LD patterns that are needed to fit multi-SNP models, we first use flashfm within each group. This estimation of joint SNP effects requires the GWAS summary statistics from each trait within each group and the group-specific LD (Fig. 1a; Methods). Therefore, the group-specific LD patterns are accounted for during the flashfm stage, and this results in trait-adjusted model PP within each group. We then obtain multi-trait multi-group model posterior probabilities by making use of the independence between the groups (Fig. 1a; Methods; *Supplementary Methods*). This differs from our flashfm multi-trait PP structure, where we account for correlations between traits in the multi-trait BF. The joint prior probability in MGflashfm also has a different form to that of flashfm. In flashfm, the joint prior is essentially the product of the marginal priors with an upweighting when there is overlap of variants between the models. The joint prior probability in MGflashfm is set under the assumption that the groups share at least one causal variant, so it is non-zero only when there is overlap between at least one pair of groups and does not include any up-weighting (Methods; *Supplementary Methods*).

In addition to the above differences in frameworks of flashfm and MGflashfm, they also differ in their final output. Flashfm leverages information between traits to output trait-specific model (and variant) PPs adjusted by the other traits. In contrast, MGflashfm uses the flashfm trait-specific PPs from each group and, for each trait, finds the multi-group model PP (mgPP) that a set of variants $C$ consists of causal variants amongst the groups; the multi-group marginal PP (mgMPP) gives the PP that a variant is causal for a subset of the groups (Fig. 1a, Methods; *Supplementary Methods*).

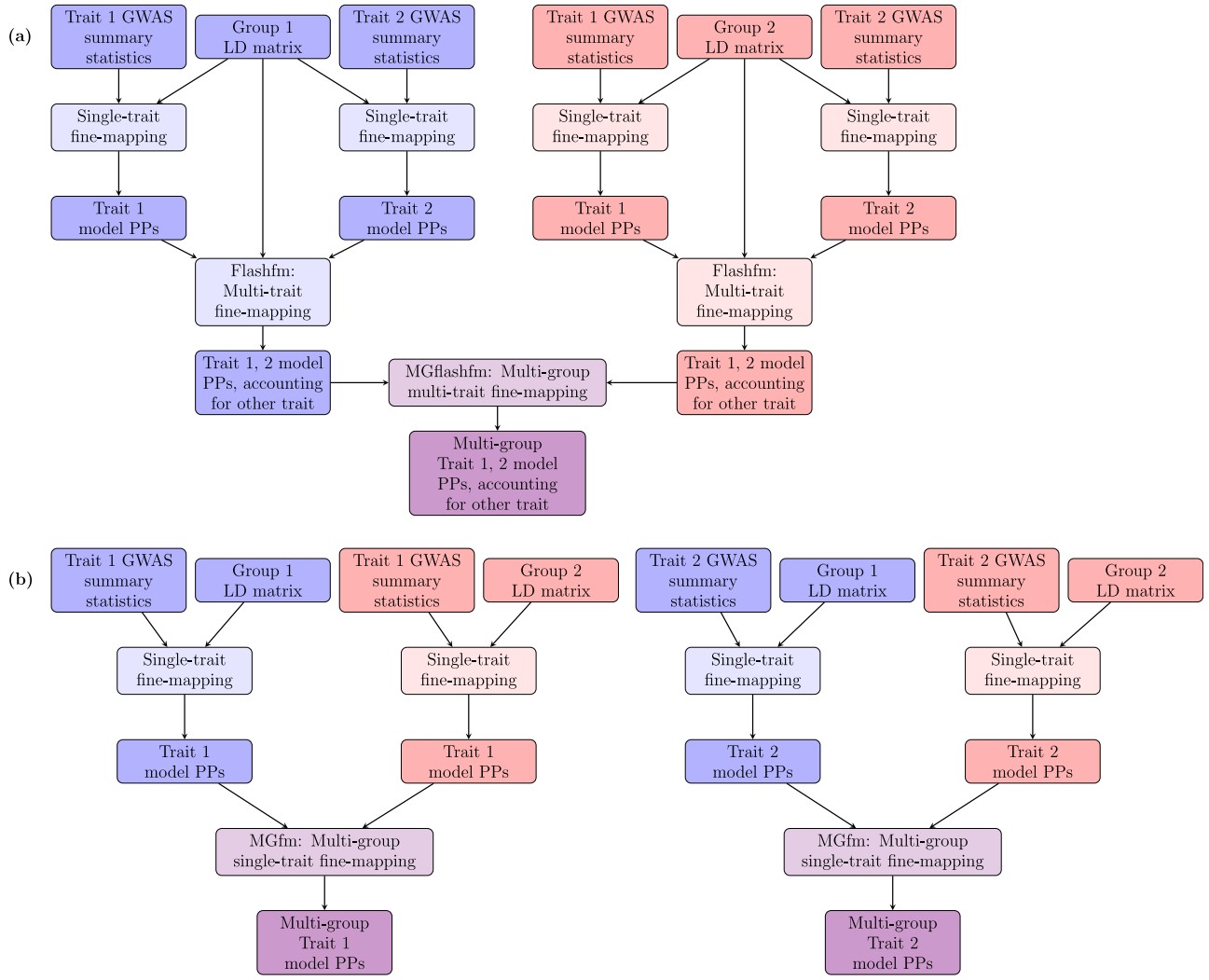

**Fig. 1 | Schematic diagrams of multi-group fine-mapping.** Diagrams are shown for two groups and two traits, and the methods are available for at most six groups and six traits. **a** In MGflashfm (multi-group multi-trait fine-mapping), multi-SNP models for each trait are first constructed within each group, using appropriate LD for the group. Within each group, multi-trait fine-mapping then leverages information between the traits while making use of group-specific LD. Trait-adjusted model PPs within each group are then jointly assessed across groups; **b** In MGfm (multi-group single-trait fine-mapping), multi-SNP models for each trait are first constructed within each group, using the group-specific LD. Then, in parallel, trait models within each group are jointly assessed across groups, independently of the other trait. For both MGfm and MGflashfm, the final output for each trait is the credible set variants, as well as the multi-group marginal PP (mgMPP) of each variant being causal, as well as other variant-specific details.

As in flashfm, rather than considering all model combinations, we reduce the model space by setting a cumulative posterior probability threshold (e.g. cpp = 0.99). For each trait within each group, we use the multi-trait (MGflashfm) or single-trait (MGfm) fine-mapping results to order the models by PP and retain those for which the sum of their PPs first passes above 0.99. These models are then assessed in multi-group fine-mapping.

Both MGflashfm and MGfm output summaries that give, for each trait, the set of variants in the credible set (99% default), and for each of these variants (i) the multi-group marginal posterior probability of the variant being causal (mgMPP); (ii) the proportion of groups that contain the variant; (iii) the names of the groups that contain the variant; and (iv) the pooled MAF among the groups that contain the variant. These details help to identify variants that may be group-specific or present in a subset of the groups. If there are no overlapping variants between the models of any pair of groups, then the joint prior is zero for all models and the message "Insufficient evidence of shared causal variants between groups." is displayed as output for that trait.

## MGflashfm, MGfm - fast, well-calibrated multi-group approaches

To assess the calibration of multi-group approaches, we use simulations to approximate the coverage at confidence level 99%—the probability that a 99% credible set (CS99) from each method will capture all causal variants (Methods). We simulated three genetically similar groups from three different populations, based on groups previously defined in 1000 Genomes[15]. As per recommendations in the National Academies Press[19], we label these groups as "African-like", "East Asian-like" and "European-like", and for brevity, we use "AFR", "EAS" and "EUR".

First, in 2-trait, 2-group simulations, we considered two sets of simulations: (i) EUR-AFR; (ii) EUR-EAS. We compared MGflashfm (multi-trait, multi-group) with three single-trait multi-group approaches: MGfm, PAINTOR[9] and msCAVIAR[10] for a region with ~330 variants. We also compare the single-group multi-trait (flashfm) results for each group, denoted (i) flashfm-EUR and flashfm-AFR; (ii) flashfm-EUR and flashfm-EAS, according to the group label used for simulations (Methods).

We also considered two-group (EUR-AFR and EUR-EAS) and three-group (EUR-EAS-AFR) simulations in a larger, more realistic-sized region of 1610 variants. For two traits over two groups, MGflashfm, MGfm and PAINTOR are relatively quick to produce results, regardless of the number of variants in a region (Supplementary Table 1). The run times for MGflashfm include sequentially running FLASHFMwithJAM (wrapper for running JAM single-trait fine-mapping (with parallelisation for traits) with flashfm) on each group, followed by the multi-group stage. Run times for MGfm include running JAM in parallel across groups, followed by the multi-group stage; as each trait is run independently, we use the maximum computational time amongst the traits. In the 1610 variant regions, the median computational times are 684 s, 395 s, and 1753 s for MGflashfm, MGfm, and PAINTOR, respectively; after 10 h, msCAVIAR was unable to run to completion. So, we do not include msCAVIAR in our comparisons for this larger, more realistic-sized region. As for MGfm, PAINTOR and msCAVIAR are not multi-trait methods, so the computational times are taken as the maximum computational time for Traits 1 and 2.

In our simulations, the causal variants for each trait are the same across groups, but causal variants are not restricted to being the same across traits; effect size for each causal variant is the same across groups. For each configuration of causal variants, we denote causal variants by upper case letters to show which variants are shared between traits and multiple letters indicate multiple causal variants for a trait. For example, Trait 1 has causal configuration AD, which indicates two causal variants A and D; this configuration for Trait 1 is the same for all groups. When there are at least two causal variants, we select one variant A such that it is common (MAF > 0.05) in AFR (for EUR-AFR and EUR-EAS-AFR) and EAS (for EUR-EAS) and low-frequency in EUR (for EUR-AFR and EUR-EAS-AFR) and EAS (for EUR-EAS-AFR) (Methods).

In both 2-group and 3-group settings, and for equal and unequal sample sizes, MGflashfm, and MGfm are consistently well-calibrated, as are independent flashfm analyses within each group (Fig. 2, Supplementary Fig. 1). PAINTOR has a lower than expected coverage in all but two settings (EUR-AFR equal sample sizes and EUR-AFR single causal variant), with a minimum coverage of 0.577 (3 groups, unequal sample sizes). Others have also highlighted low coverage of credible sets from PAINTOR[7]. When there are unequal sample sizes, msCAVIAR only meets the expected coverage when there is a single causal variant; it is only well-calibrated for multiple causal variants when there are equal sample sizes among groups.

Next, we examined the false discovery rate (FDR), defined as the mean proportion of non-causal variants having PP above a certain threshold (e.g. 0.5, 0.9), and the power, defined as the mean proportion of causal variants having PP above a certain threshold (e.g. 0.5, 0.9). For 2-trait simulations, where traits each have two causal variants, of which one is shared (T1: AD, T2: AC), we compared power and FDR within two (EUR-AFR, EUR-EAS) and three (EUR-EAS-AFR) ancestries with unequal sample sizes (90,000–10,000; 90,000–40,000–10,000). There is a general pattern of highest power for MGflashfm and MGfm, and similarly high powers for the group-specific flashfm and PAINTOR (Supplementary Fig. 2). The power and FDR for msCAVIAR are consistently 0, so not included in the plots. This is due to msCAVIAR's uniformly low PP for causal variants, with the mean PP for causal variants near 0.3. The FDR of MGflashfm, MGfm, and the group-specific flashfm are similarly low at PP threshold 0.9, but PAINTOR has very high FDR, of similar magnitude to its power (Supplementary Fig. 3).

Since PAINTOR and msCAVIAR were not well-calibrated or computationally feasible, we focus on flashfm, MGflashfm, and MGfm in the following comparisons of prioritisation and resolution. To assess prioritisation of the causal variants, we examine the distribution of the minimum MPP (marginal posterior probability that a variant is causal) of the causal variants. This indicates the frequency that both causal variants have MPP above a threshold. For example, MGflashfm is the only method that assigns MPP > 0.75 to both causal variants, in all settings (Fig. 3a, Supplementary Fig. 4a). When there are no shared causal variants (trait 1: AD, trait 2: C), MGflashfm and MGfm have very similar results—this is expected since flashfm gives similar results to single-trait fine-mapping in the absence of shared causal variants. Having a shared causal variant between traits (trait 1: AD, trait 2: AC), there is a clear improvement in prioritisation of both causal variants by MGflashfm over MGfm.

We measure resolution by considering the number of variants in the CS99 (i.e. size of sets) constructed from MGflashfm and MGfm (Methods). When Traits 1 and 2 each have two causal variants, of which one is shared, the CS99 constructed from MGflashfm are significantly smaller (significance level 0.05) than those from MGfm; one-sided paired t-tests for each trait in each simulation setting yield p-values between $7.8 \times 10^{-9}$ and $2.6 \times 10^{-3}$. For simulations where there are no shared causal variants between traits, the resolution gain of MGflashfm over MGfm is not as striking, which is expected (Fig. 3b, Supplementary Fig. 4b). With no shared causal variants, the MGflashfm CS99 are also smaller than those from MGfm for all traits and settings (all p value < 0.025), except for Trait 1 under EUR-EAS (p value = 0.5). In 3-group simulations, MGflashfm is also found to have the highest precision and resolution (Supplementary Fig. 5); for each trait and setting MGflashfm CS99 were significantly smaller than MGfm with p values between $2.2 \times 10^{-16}$ and 0.01.

Next, we include comparisons with a new multi-trait fine-mapping method, mvSUSIE[17], for each population group to confirm the advantages of multi-group multi-trait fine-mapping with MGflashfm. We simulate two traits (each with two causal variants, of which one is shared, i.e. T1: AD, T2: AC) for EUR-AFR with sample sizes of 90,000 and 10,000, using the region of 1610 variants and compare power and FDR for MGflashfm, MGfm, flashfm-EUR, flashfm-AFR, mvSUSIE-EUR, and mvSUSIE-AFR. We note that mvSUSIE returns cross-trait PIP (posterior inclusion probability) and does not return trait-specific PIP that would be analogous to the MPP of the flashfm methods. To infer which variants affect particular traits, mvSUSIE outputs the lfsr (local false sign rate) for each variant under each trait; like p-values, small values indicate an impact on the trait and we use lfsr thresholds of 0.01 and 0.1, as there is no clear mapping between the two types of thresholds. We use MPP threshold 0.9 and find that MGflashfm and MGfm have the highest power, whilst the multi-trait methods flashfm and mvSUSIE have similar power (Fig. 4). The FDR are generally similarly low for all methods, but slightly higher for mvSUSIE-EUR and lowest for flashfm-AFR (Fig. 4); there are longer LD blocks within EUR than in AFR.

The new multi-group methods, MGflashfm and MGfm are available for up to six groups; MGflashfm is available for up to six traits. Since MGfm does not include the extra step of multi-trait fine-mapping within each group, it tends to be slightly faster than MGflashfm. We also note that time is measured for MGfm by taking the maximum time between the two traits. Varying the number of groups for two traits, the median MGflashfm computational time ranges from 684 s for two groups to 1090 s for five groups, while they range from 395 s to 623 s for MGfm (Supplementary Table 2). As MGfm is run on single traits, when we vary the number of traits for two groups we only measure the computational time for MGflashfm; the MGflashfm times range from 684 s to 1214 s for two to four traits (Supplementary Table 3).

## Exclusion of causal variant retains calibration of MGflashfm

We also investigated the behaviour of MGflashfm when a causal variant is excluded from one of two groups. Consequently, this causal variant is removed from multi-group analysis with PAINTOR[5] and msCAVIAR[10], but retained in MGflashfm and MGfm. For the groups EUR and AFR, we simulate two traits, where causal variants for each trait are the same across EUR and AFR. In EUR and AFR, Trait 1 has two causal variants

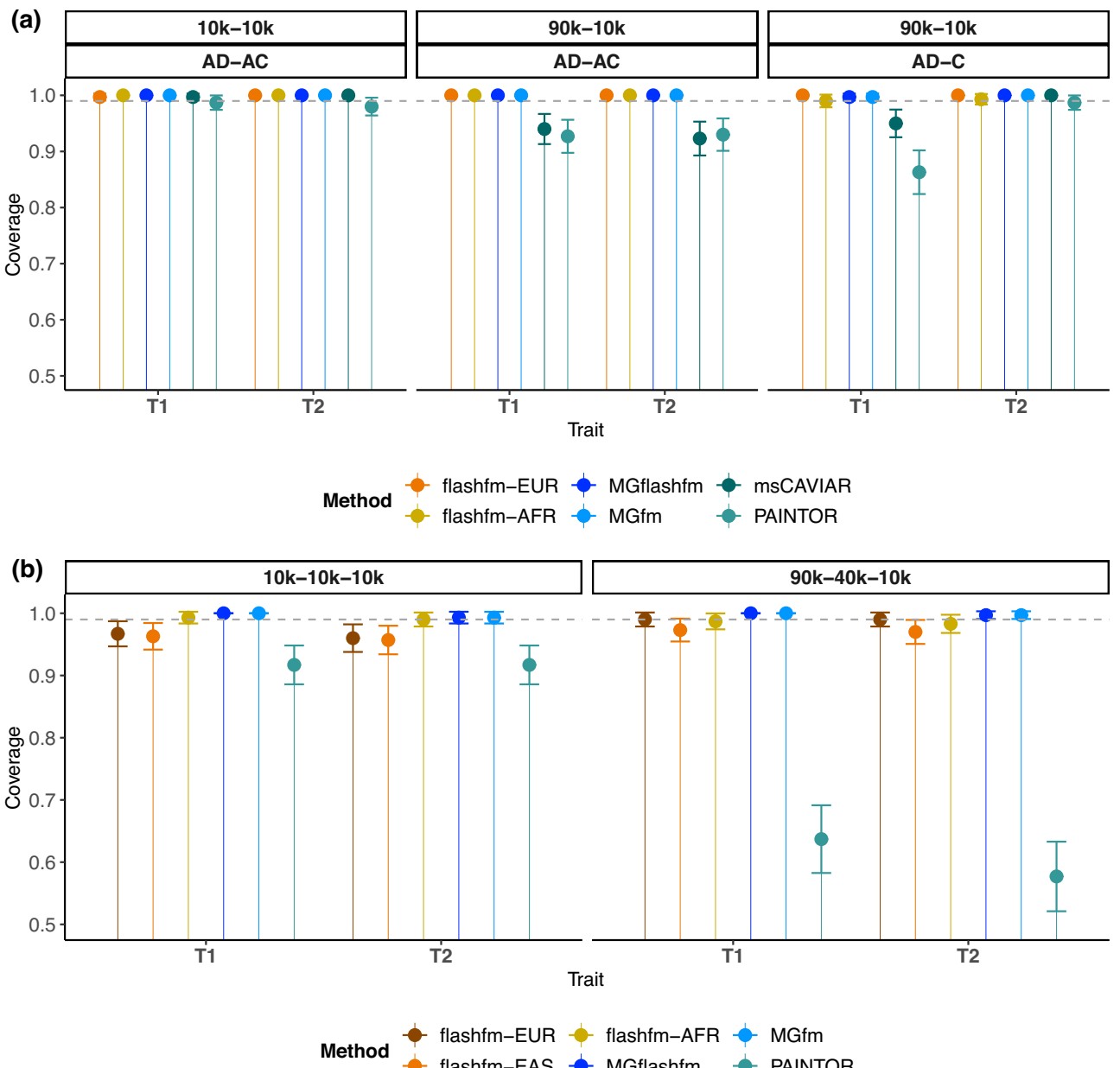

**Fig. 2 | Flashfm, MGflashfm and MGfm, are well-calibrated.** Coverage is measured as the probability that all causal variants are captured by the 99% credible set, estimated over 300 replications. Data are presented as the proportion of replications in which the 99% credible set contains all causal variants ± SEM, where SEM is the standard proportion error bound of a 95% confidence interval based on 300 observations. Flashfm-EUR, flashfm-EAS and flashfm-AFR are multi-trait (single-group) fine-mapping for the indicated group and are well-calibrated in all settings, as are MGflashfm and MGfm. PAINTOR and msCAVIAR are not well-calibrated for unequal sample sizes, though msCAVIAR is well-calibrated in the single causal variant setting. **a** Coverage results from EUR-AFR simulations. Within each panel the three simulation settings are shown as either having equal sample sizes of 10k each or sample sizes of 90k EUR and 10k AFR, and either two causal variants for each trait with one shared (trait 1: AD, trait 2: AC) or non-overlapping causal variants and one trait having a single causal variant (trait 1: AD, trait 2: C); any pair of causal variants have $r^2 < 0.5$. **b** Coverage results from EUR-EAS-AFR simulations with equal sample sizes of 10k each or 90k EUR, 40k EAS, and 10k AFR. In both settings each trait has two causal variants (trait 1: AD, trait 2: AC). The A variant has $0.005 < MAF < 0.05$ in EUR and EAS groups, but $MAF > 0.05$ in the AFR group, and the C and D variants have $MAF > 0.05$ in all groups.

labelled A and D, and Trait 2 has two causal variants A and C; the traits share one causal variant, A. We select causal variant A such that it has $MAF < 0.01$ in EUR and $MAF > 0.01$ in AFR, and assume that it fails quality control in EUR. The C and D variants are selected such that they have $MAF > 0.05$ in both EUR and AFR groups (Methods). In this setting, as expected, flashfm fine-mapping of EUR is only able to capture a single causal variant in its CS99, whereas the CS99 from flashfm of AFR is well-calibrated to capture both causal variants (Supplementary Table 4), as are MGfm and MGflashfm.

## Out-of-sample LD helps fine-map signals from GWAS summary data

It has been shown that FINEMAP, with maximum number of causal variants set to ten, tends to give multiple false positives with an out-of-sample reference panel that is "small" relative to the GWAS sample size[16]. More recently, the use of small, out-of-sample RPs in fine mapping was extensively explored in simulation studies, indicating a noticeable improvement when the true number of causal variants is specified, rather than a generous upper bound[20]. Building on the

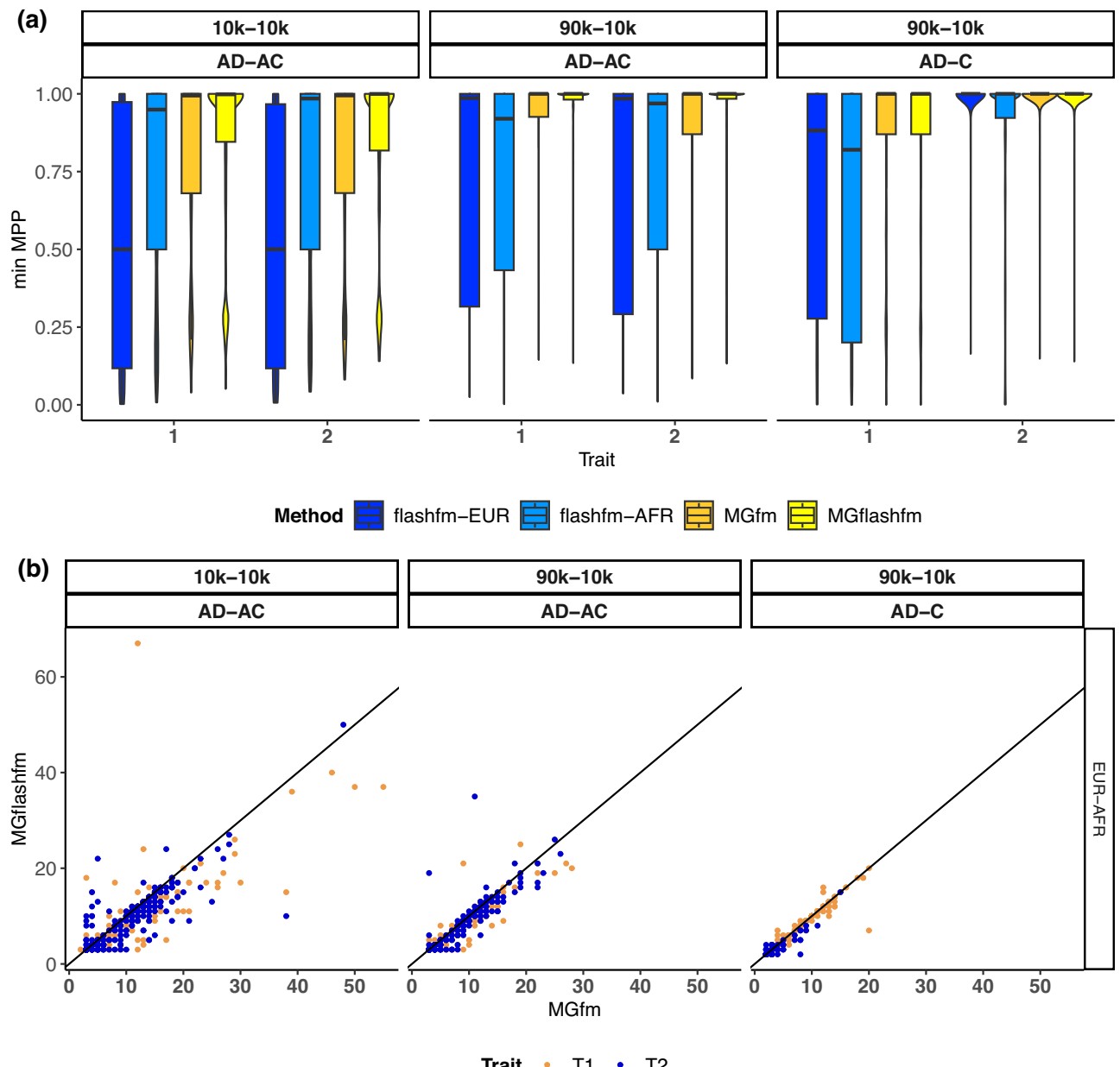

**Fig. 3 | MGflashfm has the highest gains in prioritisation and resolution among calibrated methods.** For EUR-AFR simulations, three simulation settings are shown as either having equal sample sizes of 10k each or sample sizes of 90k EUR and 10k AFR, and either two causal variants for each trait with one shared (trait 1: AD, trait 2: AC) or non-overlapping causal variants and one trait having a single causal variant (trait 1: AD, trait 2: C); any pair of causal variants have $r^2 < 0.5$ and there are 300 replications within each setting. **a** Distribution of the minimum MPP of causal variants for each trait via violin plots; the median is given by the centre line, upper and lower quartiles are the box limits, whiskers are at most 1.5× interquartile range, and width indicates the frequency. This indicates that MGflashfm is best at prioritising causal variants when the traits share a causal variant or similar performance to MGfm when no sharing. **b** Comparison of the sizes of 99% credible sets from MGflashfm and MGfm. This suggests that MGflashfm tends to have better resolution than MGfm.

simulation study results of Zou et al. [20], we propose a practical approach to fine-mapping that enables prioritisation of causal variants based on GWAS summary statistics when in-sample LD information is unavailable (Methods). We demonstrate its utility on GWAS summary statistics for four lipids traits in five groups, as made available by the Global Lipids Genetics Consortium (GLGC)[18].

GLGC performed a multi-group genome-wide meta-analysis of lipid levels in 1.65 million people. Their multi-group meta-analysis included five genetically similar groups that are labelled by continent. For consistency, we have kept the same group labels as those published[18]: admixed African or African (AFR, $N = 99,432$, 6.0% of the sample); East Asian (EAS, $N = 146,492$, 8.9%); European (EUR,

$N = 1,320,016$, 79.8%); Hispanic (HIS, $N = 48,057$, 2.9%); and South Asian (SAS, $N = 40,963$, 2.5%). We consider four of their five blood lipids traits: low-density lipoprotein cholesterol (LDL), high-density lipoprotein cholesterol (HDL), triglycerides (TG), and total cholesterol (TC); non-high-density lipoprotein cholesterol (nonHDL-C) is excluded due to its higher number of missing variants in any given region, compared to the other four traits.

In our fine-mapping of signals in the GLGC lipid traits[18] from five groups, each group with a genetically similar 1000 Genomes group as a RP, and a maximum of 10 causal variants, we observed the previously flagged pattern[16,20] (i.e. a large causal model with high PP with single-trait fine-mapping approaches, such as FINEMAP[21] and JAM[22]). To

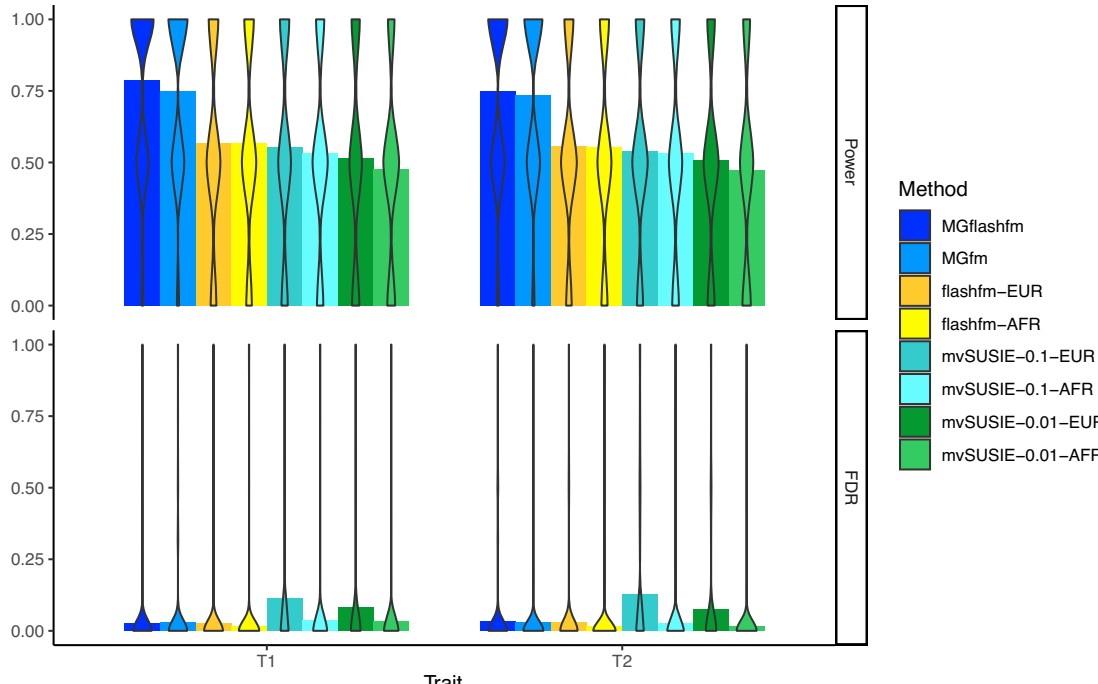

**Fig. 4 | MGflashfm has the highest power and low FDR.** For EUR-AFR simulations of two traits, results are summarised for sample sizes of 90k EUR and 10k AFR, where there are two causal variants for each trait with one shared (trait 1: AD, trait 2: AC); any pair of causal variants have $r^2 < 0.5$ and there are 300 replications within each setting. The mean power and mean FDR are shown for each method, as indicated by the top of each bar; the distribution of the power and FDR estimates over the 300 replications is shown by violin plots, where width indicates frequency. Power and FDR for the flashfm family of methods are calculated using a MPP threshold of 0.9, and for mvSUSIE lfsr thresholds of 0.1 and 0.01 are used. The power is highest for MGflashfm, followed by MGfm, then the group-specific flashfm and mvSUSIE methods. FDR is relatively low and similar amongst all methods, though lowest for flashfm-AFR and highest for mvSUSIE-EUR.

encourage parsimonious modelling of causal variants and minimise false positives that tend to occur with a generous upper limit, we have modified our extended version of JAM that is in the flashfm R package[13], so that the maximum number of causal variants is adjusted according to the data (Methods).

Using our modified JAM algorithm (JAMdynamic in our MGflashfm R package) with a dynamic upper bound for the number of causal variants, we fine-mapped signals in 50 regions that had signals for multiple traits in at least two groups, using the appropriate 1000 Genomes RP. Within each group, the favoured model (highest PP) had more than one causal variant for 49% (EUR), 17% (EAS), 11% (AFR), 6% (HIS) and 4% (SAS) of the trait-region results; 30% of the favoured EUR models had two causal variants (Fig. 5, Supplementary Data 1.1).

**Shared and distinct lipids variants prioritised across groups**

We used MGflashfm and MGfm to fine-map signals among HDL, LDL, TG, and TC in the five groups. MGflashfm tends to produce smaller CS99 than MGfm (Fig. 6, Supplementary Data 1.2). Considering the CS99 constructed for all traits, the median CS99 size reduction of MGflashfm compared to MGfm is 10.5%; by trait, the median reductions are 21.1% (TG), 12.8% (LDL), 8.16% (TC), and 7.69% (HDL). Both MGflashfm and MGfm prioritise 19 variants (mgMPP > 0.90; multi-group marginal posterior probability of a variant being causal) that are missense, stop-gained or splice variants; 63% (12/19) of these variants are not identified by any of the GLGC fine-mapping analyses (Supplementary Data 1.3). We provide details of all MGflashfm or MGfm prioritised variants, including nearest genes, functional annotations, contributing population groups, as well as whether they were also prioritised by each GLGC analysis (Supplementary Data 1.3). Among all traits, MGflashfm prioritised 185 unique variants, of which 77% are new compared to any of the GLGC analyses (EUR, AFR, multi-group); 168 variants are prioritised by MGfm (Supplementary Data 1.4).

We highlight two regions where MGflashfm, and sometimes MGfm, prioritise variants as causal, where the variant was also identified by previous fine-mapping and/or has functional annotations that increase the plausibility of it being causal. Among these regions, we also find likely causal variants that appear to be jointly causal within groups and/or variants that are non-monomorphic in a subset of groups. Such variants cannot be detected by current multi-group approaches that restrict analysis to variants appearing in all groups, nor by methods that assume a single causal variant.

In fine-mapping LDL and TC signals in 1:55405647−55605647 (*PCSK9, USP24*) MGflashfm and MGfm prioritise a missense variant, rs28362263, which is only polymorphic in AFR and HIS (pooled MAF = 0.050), and not previously prioritised by GLGC (Supplementary Data 1.3). Stop-gain variant rs28362286 is only polymorphic in AFR (MAF = 0.00798) and is prioritised by MGflashfm and MGfm, as well as the GLGC AFR analysis, but not by the GLGC multi-group analysis. Missense variant rs11591147, present only in EUR and HIS groups (pooled MAF = 0.0122), is prioritised by MGflashfm and MGfm. It is also the only variant in the GLGC CS99 for EUR and for multi-group and has been prioritised in previous fine-mapping analyses for LDL, TC, and Apolipoprotein B levels in UK Biobank[23].

In a region harbouring *APOB* (2:21131524-21331524), fine-mapping signals from the four lipids traits, MGflashfm and MGfm favour the same missense variant as prioritised by GLGC (EUR, AFR, and multi-group) for HDL: rs676210 (Supplementary Data 1.3); for HDL in UK Biobank it had PP = 0.957[23]. Only polymorphic in EAS, 3' UTR variant rs57825321 is newly prioritised for LDL and TC by MGflashfm and MGfm. In agreement with the GLGC multi-group fine-mapping effort, upstream gene variant rs934197 is prioritised for LDL by both MGflashfm and MGfm; for TC it has MGfm mgMPP = 0.348, which increases to 0.944 by leveraging information between traits in MGflashfm. Not prioritised by GLGC, missense variant rs533617 (deleterious SIFT prediction; EUR, HIS, SAS, pooled MAF = 0.0281) is

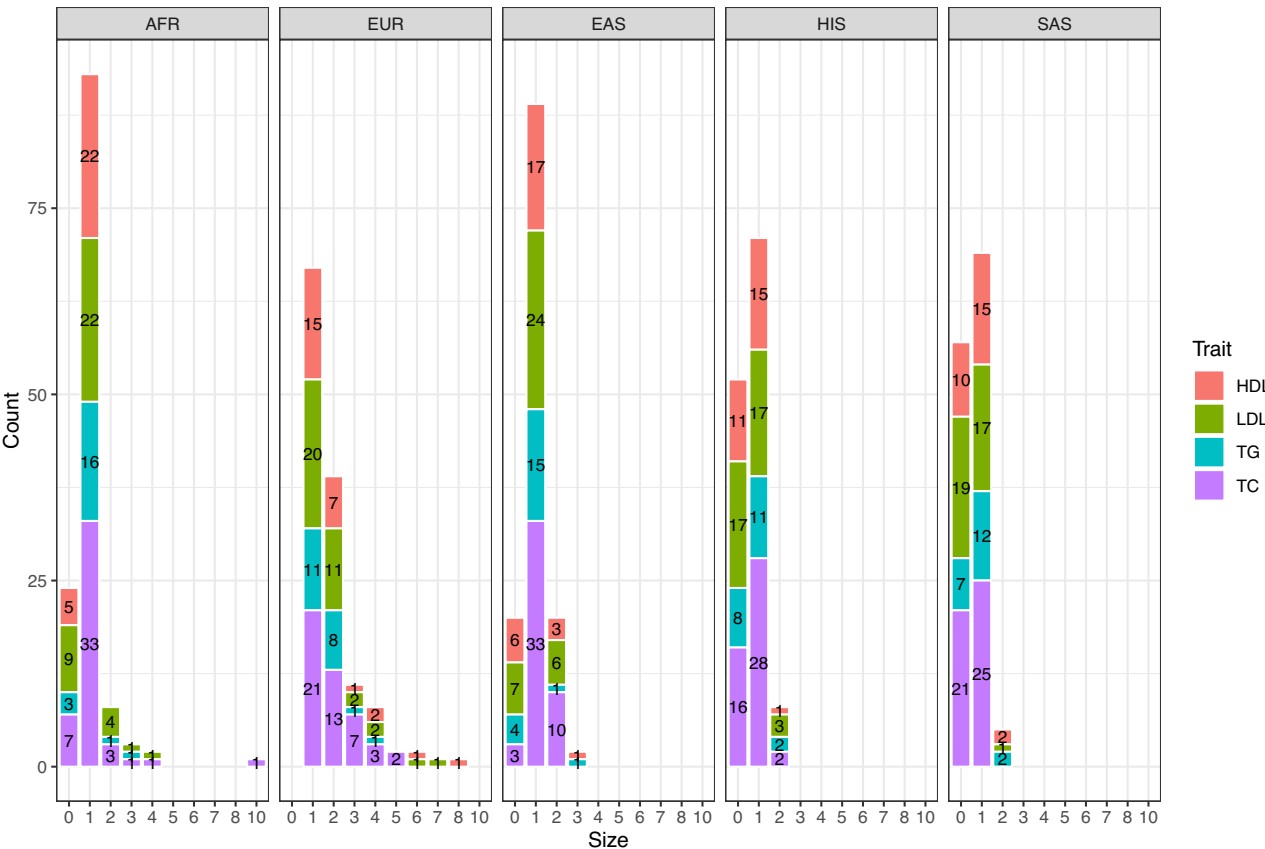

**Fig. 5 | Practical fine-mapping with 1000 Genomes reference panels favours one and two variant models.** Among 50 regions, HDL, LDL, TG and/or TC signals were fine-mapped in each of the five groups, using 1000 Genomes data (matched appropriately) and our practical approach.

prioritised by MGfm and MGflashfm and has been previously found to have PP = 1 for LDL[23,24] and for TC[23] in UK Biobank.

## Discussion

Our proposed approaches, MGflashfm and MGfm, for multi-group fine-mapping of signals among quantitative traits, offer a comprehensive approach to prioritising causal variants that are shared amongst all or a subset of groups. This removes the common restriction of multi-group methods that allow for multiple causal variants: that all variants must appear in all groups. This restriction becomes increasingly prohibitive as the number of studies and/or study diversity increases.

As with other methods that make use of GWAS summary statistics, care is needed in selecting an appropriate reference panel for LD information to model joint effects of variants within each genetically similar group. It is ideal to have access to in-sample LD, but this is often difficult to obtain when analysing previously published GWAS summary statistics. We provide a practical approach to fine-mapping with an out-of-sample LD source such as the 1000 Genomes Project[15], by taking advantage of previous extensive simulation results. Such results show that for multiple causal variant fine-mapping methods with out-of-sample LD: (i) generous upper bound of maximum number of causal variants tends to result in false positives that "fill" the available slots for variants in the model[16,20]; (ii) the correct number of causal variants as the upper bound tends to make fine-mapping results more robust[20].

In our quality control of GWAS summary statistics, within each genetically similar group, we removed any variants that were not measured in at least 80% of the individuals of that group. This was done within each group, independently of the others, since MGflashfm

and MGfm do not require the intersection of variants across groups, as for multi-group methods msCAVIAR[10] and PAINTOR[9]. It is possible that variants with allele codings A/T or G/C that have MAF near 0.5, and also an effect on the traits, may lead to unstable results. This is due to the possibility that the strand may be mismatched, but this cannot be detected because of the similarity in frequencies of both alleles. We do not remove such variants, as it is possible to have MAF near 0.5 in one group, but not other groups, and err on the side of not excluding more variants than necessary. However, there is the risk that the strand may be wrong in the group with MAF near 0.5, so caution is needed when interpreting results that prioritise such variants.

We have integrated our dynamic version of JAM[22] single-trait fine-mapping with MGflashfm and MGfm, for convenient use in wrapper functions MGFLASHFMwithJAM and MGFMwithJAM. On the MGflashfm GitHub page (https://jennasimit.github.io/MGflashfm/), we also provide guidance for how to integrate FINEMAP[21] with these multi-group methods. Our R functions are flexible, so that any single-trait fine-mapping method that outputs model PP may be integrated with flashfm[13], MGflashfm, and MGfm.

Currently the prior probabilities in MGflashfm and MGfm do not account for functional annotation, and we examine functional annotation of the credible set variants. These methods may further refine credible sets by incorporating functional annotation into the prior probabilities as has been done for PAINTOR[9] and PolyFun[25].

## Methods

### Multi-group fine-mapping framework

The first stage leverages information between traits in the same group, while accounting for group-specific LD, resulting in group-specific trait-adjusted model PPs for each trait (Fig. 1a).

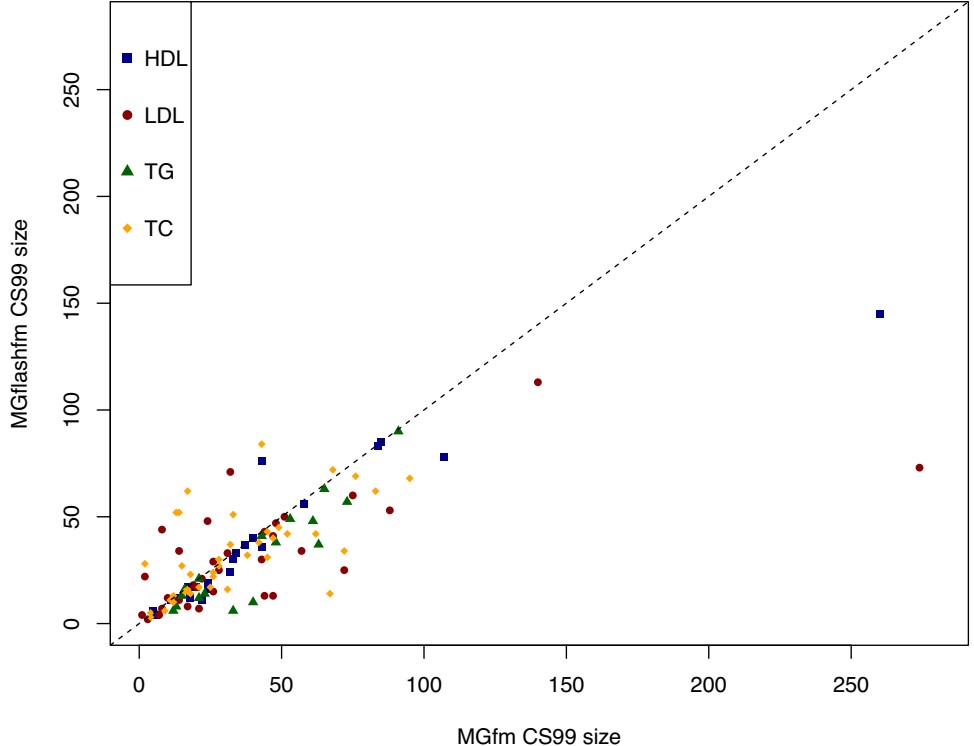

**Fig. 6 | MGflashfm generally gives smaller credible sets than MGfm for GLGC lipids.** For each of the 50 regions, the CS99 for a given trait is constructed from MGflashfm and MGfm. Most of the CS99 sizes from MGflashfm are smaller than those from MGfm.

We show (*Supplementary Methods*) that, for a particular trait, the multi-group multi-trait joint model Bayes factor (BF) is given by

$$BF_{M_1,M_2}^{(1,2)} = BF_{M_1}^{(1)} \left(\frac{N_1}{N}\right)^{\frac{m_1}{2}} BF_{M_2}^{(2)} \left(\frac{N_2}{N}\right)^{\frac{m_2}{2}} \quad (1)$$

where, for $k = 1,2$, $BF_{M_k}^{(k)}$ denotes the trait-adjusted BF for trait model $M_k$ in group $k$, $m_k$ is the number of variants in model $M_k$, $N_k$ is the sample size (or effective sample size) for the trait in group $k$, and $N = N_1 + N_2$. For group $k$, the trait-adjusted BF incorporates its group-specific LD when estimating the joint effects of multi-SNP models. In this way, the different LD structures are considered within the multi-trait stage within each group, and are then used to obtain multi-group BFs, $BF_{M_1,M_2}^{(1,2)}$.

Since the MGflashfm BF takes advantage of the independence between groups, it has broad similarity to mcoloc[12] which assumes independent traits and sets the joint BF as the product of the marginal trait BFs. However, mcoloc builds on the Wakefield approximate BF[26], which assumes a single causal variant and requires specification of the variance of the normal prior for the effect estimates. In contrast, flashfm and MGflashfm build on the Bayesian Information Criterion BF approximation, which allows multiple causal variants[27]. For example, in the derivation of flashfm, we show that, without loss of generality, assuming mean 0 for a single trait $y$, the log(BF) for a model $M$ with $m$ causal variants $\mathbf{X}$ has the form

$$log(ABF_M) = -\frac{N}{2}\left(\frac{(y - X\hat{\beta})^T(y - X\hat{\beta})}{y^T y}\right) - \frac{m}{2} log(N) \quad (2)$$

where $\hat{\beta}$ is the vector of maximum likelihood estimates for the effects.

We note that in our calculation of the model BFs within each group, the GWAS summary statistics and group-specific LD are used when estimating the joint effects in the multi-SNP models for each

trait. As these BF estimates depend on the GWAS summary statistics within each group, they will be correct at the same MAF thresholds for inclusion of variants in the GWAS. In general, for larger samples (≥10,000) we recommend using this approximation for variants with MAF ≥ 0.001, and for smaller samples to use MAF ≥ 0.005.

At the multi-group stage, the prior probability for a joint multi-group model of one trait is different from the flashfm prior used for a joint multi-trait model of a single group, where priors are up-weighted when variants between models overlap. In the 2-group setting, a natural joint prior probability for an $m_1$-SNP group 1 model $M_1$ with an $m_2$-SNP group 2 model $M_2$ is $p_{m_1,m_2}^{(1,2)} = p_{m_1} p_{m_2}$, where $p_{m_1}$ and $p_{m_2}$ are prior probabilities of an $m_1$-SNP model and $m_2$-SNP model, respectively. Assuming that the groups share at least one causal variant, we add the restriction that the joint prior is only non-zero when the models overlap, so that the multi-group joint prior is

$$p_{m_1,m_2}^{(1,2)} = p_{m_1} p_{m_2} \mathbf{1}\{M_1 \cap M_2 \neq \varnothing\} \tau_{m_1,m_2} \quad (3)$$

where $\tau_{m_1,m_2}$ is a correction factor given by

$$\tau_{m_1,m_2} = \frac{\binom{n}{m_2}}{\binom{n}{m_2} - \binom{n - m_1}{m_2}} \quad (4)$$

with $n$ being the total number of variants in the region; $\tau_{m_1,m_2}$ ensures that the total joint prior probability of an $m_1$-SNP group 1 model with an $m_2$-SNP group 2 model in the reduced search space is anchored to remain the same as in the full model search space (*Supplementary Methods*). For three groups, the multi-group joint prior is

$$p_{m_1m_2m_3}^{(1,2,3)} = p_{m_1} p_{m_2} p_{m_3} \mathbf{1}\{M_1 \cap M_2 \neq \varnothing \text{ or } M_1 \cap M_3 \neq \varnothing \text{ or } M_2 \cap M_3$$
$$\neq \varnothing\} \tau_{m_1m_2m_3} \quad (5)$$

With combinatorial arguments used to derive $\tau_{m_1 m_2 m_3}$ and this is extended in a similar manner for four, five and six groups.

It follows that the multi-group multi-trait joint model PP for two groups is given by

$$PP^{(1,2)}_{M_1,M_2} = PP^{(1)}_{M_1}\left(\frac{N_1}{N}\right)^{\frac{m_1}{2}} PP^{(2)}_{M_2}\left(\frac{N_2}{N}\right)^{\frac{m_2}{2}} 1\{M_1 \cap M_2 \neq \varnothing\}\tau_{m_1,m_2} \quad (6)$$

where $PP^{(1)}_{M_1}$ is the flashfm trait-adjusted PP for model $M_1$ in group 1 and $PP^{(2)}_{M_2}$ is the flashfm trait-adjusted PP for model $M_2$ in group 2. It is similarly defined for more than two groups (*Supplementary Methods*).

We also use this same framework for MGfm by setting $PP^{(1)}_{M_1}$ as the single-trait PP for model $M_1$ in group 1 and $PP^{(2)}_{M_2}$ as the single-trait PP for model $M_2$ in group 2 (Fig. 1b).

In flashfm, we obtain marginal model PPs for each trait by fixing each trait and summing over the joint trait PPs for that trait. For example, when there are two traits, the PP for model $M_i$ of trait 1 is the sum over all joint PP for each of the models for trait 2 with model $M_i$ for trait 1. For MGflashfm, after obtaining the multi-group multi-trait joint model PP $PP^{(1,2)}_{M_1 M_2}$, as above, we calculate the multi-group model PP (mgPP) for each trait that gives the PP for a set of variants $C$, consisting of variants in a multi-group model. This encompasses all group 1-group 2 models that share at least one variant and $C$ is the collection of all variants in these models. So, the mgPP for a set $C$ is the sum over all combinations of group 1 and group 2 models that share at least one variant and the union of variants is $C$ (*Supplementary Methods*):

$$mgPP_C = \sum_{i,j \in S} PP^{(1,2)}_{ij}; S = \{(i,j): M^{(1)}_i \cup M^{(2)}_j = C, M^{(1)}_i \cap M^{(2)}_j \neq \varnothing\} \quad (7)$$

Finally, for variant $s$, multi-group MPPs are found from

$$mgMPP_s = \sum_{C:s \in C} PP_C \quad (8)$$

## Multi-trait simulations for multiple groups

We selected *APOE*, at 19:45300000-45500000, as a focus region for our simulations because it contains several genetic associations for cardiometabolic related traits. Haplotypes were simulated with HAPGEN2[28] using 1000 Genomes Phase 3 reference panels[15]: CEU + TSI for European (EUR), LWK + YRI for African (AFR), and CHB + JPT for East Asian (EAS) genetically similar groups. In our simulations that involve comparisons of msCAVIAR, we reduce our region to 19:45386029-45439498; msCAVIAR was unable to run to completion for the larger region within 10 h.

In our 2-group simulations we consider both EUR-AFR and EUR-EAS settings with equal sample sizes between groups (as a baseline comparison) and the more realistic setting where the EUR group has a larger sample size of 90,000 and the second group has 10,000 individuals. In 3-group simulations, we set sample sizes of 90,000, 40,000 and 10,000, for EUR, EAS and AFR groups, respectively.

We simulated between one and three causal variants for each trait, and in some settings traits shared some (not all) causal variant(s). In 2-group simulations, traits with at least two causal variants had one variant that was randomly selected such that it had $0.001 < \text{MAF} < 0.05$ in EUR and MAF > 0.05 in AFR/EAS; for 3-group simulations it had $0.001 < \text{MAF} < 0.05$ in both EUR and EAS groups and MAF > 0.05 in the AFR group. All other causal variants were randomly selected such that they had MAF > 0.05 in all groups included in the simulation. Within each group, any pair of causal variants satisfied $r^2 < 0.5$ and each common variant (MAF > 0.05) had $r^2 > 0.6$ with at least six other variants in the region. Causal effects were randomly selected from a Uniform distribution with minimum 0.01 and maximum 0.6 and each variant had the same effect size across groups.

In 2-group simulations for two traits with two causal variants, of which one is shared, we also compare all flashfm methods with the

multi-trait method, mvSUSIE[17]. In our implementation of mvSUSIE we used the canonical prior and followed the author's suggestion of estimating the residual variance using the variants with absolute Z score below 2 for all traits; we also set coverage to 0.99.

For $M$ traits, the measurement for trait $k$ of individual $j$, $y_{kj}$, is obtained from

$$y_{kj} = \sum_{i=1}^{m_k} \beta_{ik} x_{ij} + \varepsilon_{kj} \quad (9)$$

where $x_{ij}$ is the number of alternative alleles of variant $i$ for individual $j$ (i.e. genotype score), $\beta_{ik}$ is the effect of causal variant $i$ for trait $k$, $m_k$ is the number of causal variants for trait $k$, and $\varepsilon_{kj}$ is the $k$th element of the $j$th multivariate Normal distributed error variable with mean **0** and covariance $\Sigma$, which is the covariance matrix of the $M$ traits; we set each trait variance to 0.2 and each trait correlation to 0.4 (covariance is 0.08).

All variants that appear in each group with MAF > 0.001 are carried forward for analysis by PAINTOR and msCAVIAR. Variants that appear with MAF > 0.001 in at least one group are considered in MGfm and MGflashfm.

We also investigate the setting where a causal variant is excluded from analysis, possibly due to the variant not passing quality control in one of the groups. In two-group (EUR-AFR) two-trait simulations we use similar settings as above, where the two traits share a causal variant A and each has a distinct second causal variant C or D. The C and D variants are selected such that they have MAF > 0.05 in both EUR and AFR groups. The A variant is selected to have MAF < 0.01 in EUR and MAF > 0.01 in AFR. After simulating the traits from the causal variants, as above, the A causal variant is removed from the EUR group, before multi-group fine-mapping with MGfm and MGflashfm, assuming that it failed quality control in the EUR group. We examined the setting where the AFR sample size is 10,000 and the EUR sample size is 90,000 and used 200 replications.

## Credible sets and calibration

For each trait, we construct 99% multi-group credible sets. First, models are sorted by decreasing PP and then models are selected from the sorted list until the cumulative sum of their PPs first exceeds 0.99. The unique variants from the selected models form the 99% credible set. As multi-group model PPs are evaluated for each configuration of trait models that have at least one overlapping variant, these credible sets tend to contain variants with high PP, but also variants with very low PP that can be viewed as noise. For this reason, credible sets from MGflashfm and MGfm tend to be larger than those from flashfm (multi-trait single-group fine-mapping), and are not comparable to them; only MGflashfm and MGfm credible set sizes are compared to each other. Instead, evidence of causality (MPP) for causal variants is compared between methods, as well as power and FDR.

To assess calibration of each method, a 99% credible set is constructed for each of the 300 replications in a simulation setting. Coverage is defined as the probability that all causal variants are contained within a credible set; a 99% credible set has expected coverage 0.99. Therefore, for each trait, we estimate the coverage by the proportion of simulation replications in which the 99% credible set contains all causal variants of the trait.

## Practical fine-mapping for broader use with out-of-sample LD

Rather than starting at a guess for the number of causal variants, the extended JAM algorithm first considers only single-SNP models and the null model. If the result is a single model with PP = 1 or no convergence, then the upper bound on causal variants is incremented to two. This gradual increment of the upper bound is continued until there is convergence of results and there is not a single model that carries all of the PP support. If the RP is a poor match for the GWAS data, this incremental procedure could continue to an unreasonably

large number, so we set a default maximum possible upper bound of 15, that users may adjust to their prior knowledge. This algorithm is built into the updated FLASHFMwithJAMd function in the MGflashfm R package; it is available for single-trait fine-mapping in the function JAMdynamic, and called within both MGFMwithJAM and MGFLASHFMwithJAM.

When the RP is from in-sample data, we recommend changing the default start value for the maximum number of causal variants from 1 to 10, or to use the original flashfm algorithm in the flashfm R package; FLASHFMwithJAM is robust to a generous upper bound on the maximum number of causal variants, so we recommend to keep this flexibility, when possible.

### Fine-mapping associations in four lipids traits in five groups

Lipids GWAS summary statistics from five genetically similar groups have been made available by the Global Lipids Genetics Consortium (GLGC), after a multi-group genome-wide meta-analysis of lipid levels in 1.65 million people[18]. Their multi-group GWAS included the following five genetically similar groups, with the following predefined labels: admixed African or African (AFR, $N = 99,432$, 6.0% of the sample); East Asian (EAS, $N = 146,492$, 8.9%); European (EUR, $N = 1,320,016$, 79.8%); Hispanic (HIS, $N = 48,057$, 2.9%); and South Asian (SAS, $N = 40,963$, 2.5%). We consider four of their five blood lipids traits: low-density lipoprotein cholesterol (LDL), high-density lipoprotein cholesterol (HDL), triglycerides (TG), and total cholesterol (TC); non-high-density lipoprotein cholesterol (nonHDL-C) is excluded due to its higher number of missing variants in any given region, compared to the other four traits.

For each group, we approximated the trait genetic correlation between each pair of lipids. This was done by constructing LD scores[29] from the most genetically similar 1000 Genomes, phase 3 reference panel[15] and using these LD scores with the group-specific GWAS summary statistics in a robust genetic correlation approximation approach[30].

We constructed 50 regions for multi-group multi-trait fine-mapping, selected based on group-specific GWAS results from the GLGC analysis[18]. In particular, we highlighted loci where multiple traits had signals in multiple groups. The start point of our region was taken as the minimum base-pair position of the index variants shifted downwards by 100 kb and the end point was taken as the maximum base-pair position of the index variants shifted upwards by 100 kb; this resulted in regions with lengths varying from 200 kb to nearly 400 kb.

Before fine-mapping, we excluded all non-biallelic variants from each group and within each group we only retained variants with MAF > 0.005. We selected a trait for multi-group fine-mapping if it had a genome-wide significant signal (i.e. $p < 5 \times 10^{-8}$) in at least two groups. As our primary focus is multi-trait multi-group fine-mapping, only regions with at least two traits that pass this criteria are carried forward for fine-mapping. Among the traits that passed our criteria for fine-mapping, within each group we kept variants that were in at least 80% of that group's individuals for all selected traits. The effect alleles in the GWAS of each group were compared to the closest matching 1000 Genomes data, and flipped if necessary; if the alleles did not match and flipping was not possible, the variant was removed.

Within each group, we first ran flashfm with JAM using the 1000 Genomes super-populations as reference panels: AFR ($N = 661$), EAS ($N = 504$), EUR ($N = 503$), SAS ($N = 489$), HIS ($N = 347$). We then proceeded with MGfm and MGflashfm. We provide the most severe consequence of variants that have mgMPP>0.90 by either method, using the Ensembl Variant Effect Predictor (VEP)[31].

### Reporting summary

Further information on research design is available in the Nature Portfolio Reporting Summary linked to this article.

## Data availability

The GLGC lipids traits GWAS summary statistics from five genetically similar groups are freely available from http://csg.sph.umich.edu/willer/public/glgc-lipids2021/results/ancestry_specific/. Reference panels for LD and LD scores were generated from the 1000 Genomes data available at https://ctg.cncr.nl/software/MAGMA/ref_data/. The detailed data results of our multi-group multi-trait fine-mapping GLGC results are given in Supplementary Data 1. For ease of access, they are also deposited in a FigShare public data repository (https://doi.org/10.6084/m9.figshare.23266703[32]). Positions are given according to hg19/build 37.

## Code availability

Our proposed multi-group fine-mapping methods, MGflashfm and MGfm, are freely available as an R library at https://jennasimit.github.io/MGflashfm/ (https://doi.org/10.5281/zenodo.7974535[33]). This library also includes updated versions of expanded JAM and flashfm that have dynamic selection of the maximum number of causal variants, as learned from the data. Custom code for the analysis of the GLGC data is available at https://github.com/fz-cambridge/MGflashfm-GLGC-analysis (https://doi.org/10.5281/zenodo.10034536[34]). Trait genetic correlations were estimated using LD scores (v1.0.1, https://github.com/bulik/ldsc) together with MTAR (http://www.github.com/baolinwu/MTAR). We simulated genotype data with hapgen2 (http://mathgen.stats.ox.ac.uk/genetics_software/hapgen/hapgen2.html). The annotation tool we used is Ensembl VEP GRCh37 (https://grch37.ensembl.org/info/docs/tools/vep/index.html).

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

## Acknowledgements

J.A. and F.Z. are supported by the UK Medical Research Council (MR/R021368/1 (JA), MC_UU_00002/4). T.C. is an international training fellow supported by the Wellcome Trust grant (214205/Z/18/Z). A.M. is supported by the UK Medical Research Council (MR/W029626/1). This work was funded in part by an "Expanding excellence in England" award from Research England to I.B. This study was supported by the National Institute for Health and Care Research Exeter Biomedical Research Centre. The views expressed are those of the author(s) and not necessarily those of the NIHR or the Department of Health and Social Care. For the purpose of Open Access, the authors have applied a CC BY public copyright licence to any Author Accepted Manuscript version arising from this submission.

## Author contributions

J.A. conceived and co-ordinated the project. J.A. developed the MGflashfm method with input from A.M. and I.B. J.A. and F.Z. developed the MGflashfm software. F.Z., O.S. and J.A. performed statistical analyses. F.Z., O.S., T.C., S.F., I.B., A.M. and J.A. contributed to interpreting results. F.Z., O.S., and J.A. wrote the first draft of the paper with input from A.M., I.B., T.C., and S.F. All authors approved the final version of the manuscript.

## Competing interests

The authors declare no competing interests.
