## [Peer Review File · Nature Communications]

Leveraging information between multiple population groups and traits improves fine-mapping resolutionREVIEWER COMMENTS

Reviewer #1 (Remarks to the Author):

The authors extend their previous method flashfm which performs multi-trait fine-mapping to cases where the LD between groups of studies are different. This can have an important impact on fine-mapping utilizing multi-ancestry GWAS.

Major comments:

1. There exists a strong connection between fine-mapping and colocalization. The authors need to make the connection and difference exist between both methods.
2. The method to multiply the multiple Bayes Factors has a lot of similarities with the mcoloc method. The authors need to compare the computed BFs and acknowledge the mcoloc method.
3. The authors need to compare their method with mvSuiSe (<https://stephenslab.github.io/mvsusieR/reference/mvsusie.html>) and CARMA (<https://www.nature.com/articles/s41588-023-01392-0>).
4. It is not clear to the reader how the model incorporates different LDs given the provided BF does not utilize LD. The authors need to elaborate in the result and provide detail in the method section.
5. It has been shown by many methods that both msCAVIAR and PAINTOR are pretty calibrated. The authors need to provide some intuition why these methods perform so badly in their simulation.
6. The main difference between MGflashfm and flashfm on the method part is not very clear. I recommend providing more detail in the overview of the method and method section.

Minor comments:

1. Can the author provide some intuition if the BF works under cases $MF < 0.05$ and in which MAF the BF approximation is not correct?
2. I was under the impression that BF approximation works under the one causal variant but authors have shown that their method works under 2 causal variants. Can they provide some intuition how the BF approximation works under more than one causal variant?

Reviewer #2 (Remarks to the Author):

In the manuscript "Leveraging information between multiple population groups and traits improves fine-mapping resolution" Zhou et al. improve fine-mapping of variants by jointly analyzing all population groups. The authors put a lot of effort in investigating the performance of the proposed method(s). We commend them for this job while raising one major and two moderately important points.

Major points:

- 1) The authors seem concerned largely only with limiting false negatives (coverage statistics). However, for the best calibration possible there is a need to also concern ourselves with false positives. Consequently, the authors should also provide the comparison of false positive rates, e.g., what fraction of # of causal SNP do the false signals that are selected by methods.

Moderate points:

- 1) Figure 1 is hard to read. You should find a way to get the text inside the text boxes to be larger

2) Edit the paper to improve readability further, e.g., eliminate some duplicated words within sentences/adjacent sentences.

RE: **NCOMMS-23-23905A**

We appreciate the constructive comments from the reviewers and feel that our revised version is an improvement over our original manuscript, thanks to their comments. All comments are addressed individually, and we list some of the significant changes here.

Within our simulation comparisons with multi-group methods PAINTOR and msCAVIAR, we have added in assessments of power and false discovery rate (FDR), which show a high FDR for PAINTOR and low power for msCAVIAR. We have also extended our simulation study to include comparisons with a new multi-trait fine-mapping method, mvSUSIE, which is designed for a single population group. We find that our multi-group multi-trait approach MGflashfm has the highest power over all methods, and a similarly low FDR to the single group flashfm and mvSUSIE approaches. We have also added in more detail on the differences between flashfm and MGflashfm, which includes construction of the joint Bayes' factors, different joint priors, and final output, which is defined in a very different way.

REVIEWER COMMENTS

Reviewer #1

The authors extend their previous method flashfm which performs multi-trait fine-mapping to cases where the LD between groups of studies are different. This can have an important impact on fine-mapping utilizing multi-ancestry GWAS.

Thank you for all of your comments that have helped to clarify and improve readability of this paper.

Major comments:

1. There exists a strong connection between fine-mapping and colocalization. The authors need to make the connection and difference exist between both methods.

We thank the reviewer for their comment. We agree that fine-mapping and colocalization approaches are somewhat related and it is important to ensure readers are clear on the similarities and differences between these two types of approach. To address this we have edited the manuscript to say:

“When multiple traits have signals in the same region, colocalisation is often used to assess the evidence that two (or more) traits share the same causal variant (and sometimes identifies the variant). However, fine-mapping aims to identify causal variants, which may or may not be shared across traits, but it takes advantage of the correlation among traits to improve localisation. Colocalisation methods often make simplifying assumptions of at most one causal variant and uncorrelated traits. For example, mcoloc¹² does not consider correlations between traits because it requires that all traits are measured in distinct datasets of unrelated individuals. In addition, colocalisation methods are designed for only a single population group.”

2. The method to multiply the multiple Bayes Factors has a lot of similarities with the mcoloc method. The authors need to compare the computed BFs and acknowledge the mcoloc method. *Thank you for this comment and we have added a clarification to distinguish between the BF calculations of the methods. Both mcoloc and MGflashfm calculate joint Bayes' factors, but each has different underlying assumptions and are based on different approximations. We have added the following to the Methods section:*

“Since the MGflashfm BF takes advantage of the independence between groups, it has broad similarity to mcoloc¹² which assumes independent traits and sets the joint BF as the product of the marginal trait BFs. However, mcoloc builds on the Wakefield approximate BF²⁶, which assumes a single causal variant and requires specification of the variance of the normal prior for the effect estimates. In contrast, flashfm and MGflashfm build on the Bayesian Information Criterion (BIC) BF approximation, which allows multiple causal variants²⁷. For example, in the derivation of flashfm, we show that, without loss of generality, assuming mean 0 for a single trait y , the $\log(\text{BF})$ for a model M with m causal variants \mathbf{X} has the form $\log(\text{BF}_M) = -\frac{N}{2} \left(\frac{(y - \mathbf{X}\hat{\beta})^T (y - \mathbf{X}\hat{\beta})}{y^T y} \right) - \frac{m}{2} \log(N)$, where $\hat{\beta}$ is the vector of maximum likelihood estimates (MLE) for the effects. ”

3. The authors need to compare their method with mvSuiSe (<https://stephenslab.github.io/mvsusieR/reference/mvsusie.html>) and CARMA (<https://www.nature.com/articles/s41588-023-01392-0>).

Thank you for this suggestion. The key novel aspect of our approach is to perform multi-trait fine-mapping across multiple different world populations, whereas neither mvSUSIE nor CARMA are designed for multiple LD matrices. We have added in simulation comparisons with mvSUSIE, which is a multi-trait method and is comparable to flashfm. However, CARMA is a single-trait fine-mapping method, which is comparable to FINEMAP, JAM, and SUSIE and not designed for multi-trait nor multi-group analyses, so we have not included it in comparisons.

We have added the following to Results, as well as a new figure (Fig 4):

“Next, we include comparisons with a new multi-trait fine-mapping method, mvSUSIE¹⁷, for each population group to confirm the advantages of multi-group multi-trait fine-mapping with MGflashfm. We simulate two traits (each with two causal variants, of which one is shared, i.e. T1: AD, T2: AC) for EUR-AFR with sample sizes of 90,000 and 10,000, using the region of 1610 variants and compare power and FDR for MGflashfm, MGfm, flashfm-EUR, flashfm-AFR, mvSUSIE-EUR, and mvSUSIE-AFR. We note that mvSUSIE returns cross-trait PIP (posterior inclusion probability) and does not return trait-specific PIP that would be analogous to the MPP of the flashfm methods. To infer which variants affect particular traits, mvSUSIE outputs the lfsr (local false sign rate) for each variant under each trait; like p-values, small values indicate an impact on the trait and we use lfsr thresholds of 0.01 and 0.1, as there is no clear mapping between the two types of thresholds. We use MPP threshold 0.9 and find that MGflashfm and MGfm have the highest power, whilst the multi-trait methods flashfm and mvSUSIE have similar power (Figure 4). The FDR are generally similarly low for all methods, but slightly higher for

mvSUSIE-EUR and lowest for flashfm-AFR (Figure 4); there are longer LD blocks within EUR than in AFR.”

We have added the following to Methods:

“In 2-group simulations for two traits with two causal variants, of which one is shared, we also compare all flashfm methods with the multi-trait method, mvSUSIE. In our implementation of mvSUSIE we used the canonical prior and followed the author’s suggestion of estimating the residual variance using the variants with absolute Z score below 2 for all traits; we also set coverage to 0.99.”

4. It is not clear to the reader how the model incorporates different LDs given the provided BF does not utilize LD. The authors need to elaborate in the result and provide detail in the method section.

Thank you for this comment that clarifies the MGflashfm framework. Since we estimate the BF for multi-SNP models, when we estimate the trait-adjusted BFs within flashfm for each group, we use the group-specific LD to estimate the joint effects.

We have added clarification to the Methods by adding the following after defining the multi-group BF:

“For group k , the trait-adjusted BF incorporates its group-specific LD when estimating the joint effects of multi-SNP models. In this way, the different LD structures are considered within the multi-trait stage within each group, and are then used to obtain multi-group BFs, $BF_{M_1, M_2}^{(1,2)}$.

Since the MGflashfm BF takes advantage of the independence between groups, it has broad similarity to mcoloc¹² which assumes independent traits and sets the joint BF as the product of the marginal trait BFs. However, mcoloc builds on the Wakefield approximate BF²⁶, which assumes a single causal variant and requires specification of the variance of the normal prior for the effect estimates. In contrast, flashfm and MGflashfm build on the Bayesian Information Criterion (BIC) BF approximation, which allows multiple causal variants²⁷. For example, in the derivation of flashfm, we show that, without loss of generality, assuming mean 0 for a single trait y , the $\log(\text{BF})$ for a model M with m causal variants \mathbf{X} has the form $\log(\text{BF}_M) =$

$-\frac{N}{2} \left(\frac{(y - \mathbf{X}\hat{\beta})^T (y - \mathbf{X}\hat{\beta})}{y^T y} \right) - \frac{m}{2} \log(N)$, where $\hat{\beta}$ is the vector of maximum likelihood estimates (MLE) for the effects.

We note that in our calculation of the model BFs within each group, the GWAS summary statistics and group-specific LD are used when estimating the joint effects in the multi-SNP models for each trait.”

And have modified the text in the methods overview of the Results to:

*“To account for the group-specific LD patterns that are needed to fit multi-SNP models, we first use flashfm within each group. This estimation of joint SNP effects requires the GWAS summary statistics from each trait within each group and the group-specific LD (**Figure 1a; Methods**). Therefore, the group-specific LD patterns are accounted for during the flashfm stage, and this results in trait-adjusted model PP within each group.”*

5. It has been shown by many methods that both msCAVIAR and PAINTOR are pretty calibrated. The authors need to provide some intuition why these methods perform so badly in their simulation.

The calibration for PAINTOR appears to vary and there are other examples in the literature where it did not perform as well in simulations. We have added the following to our Results section (before Figure 2).

“Others have also highlighted low coverage of credible sets from PAINTOR⁸.”

There is some miscalibration for msCAVIAR, but its computational feasibility was an impediment to its inclusion in further simulations of more realistic-sized regions. In addition, we now include power and FDR comparisons, where we find that both power and FDR are zero for msCAVIAR; the causal variants have average PP near 0.3, which explains its low power.

6. The main difference between MGflashfm and flashfm on the method part is not very clear. I recommend providing more detail in the overview of the method and method section.

Thank you for this important point. MGflashfm and flashfm differ in the following ways (i) construction of the joint BFs, where trait correlations are accounted for in the joint trait BFs, but the joint group BFs make use of the independence between groups; (ii) different joint priors, where flashfm gives upweighting when there is overlap between trait models, whilst MGflashfm does not use upweighting in the multi-group prior, which is non-zero only when models overlap between at least one pair of groups; (iii) final output, where flashfm gives trait-specific model (and variant) PPs, whilst by nature, MGflashfm gives multi-group (not group-specific) model and variant (PPs), which are defined in a very different way to flashfm (details given below).

We have clarified the differences by adding the following to the overview in the Results section:

“This differs from our flashfm multi-trait PP structure, where we account for correlations between traits in the multi-trait BF. The joint prior probability in MGflashfm also has a different form to that of flashfm. In flashfm, the joint prior is essentially the product of the marginal priors with an upweighting when there is overlap of variants between the models. The joint prior probability in MGflashfm is set under the assumption that the groups share at least one causal variant, so it is non-zero only when there is overlap between at least one pair of groups and does not include any up-weighting (Methods; Supplementary Methods).

In addition to the above differences in frameworks of flashfm and MGflashfm, they also differ in their final output. Flashfm leverages information between traits to output trait-specific model (and variant) PPs adjusted by the other traits. In contrast, MGflashfm uses the flashfm trait-specific PPs from each group and, for each trait, finds the multi-group model PP (mgPP) that a set of variants C consists of causal variants amongst the groups; the multi-group marginal PP (mgMPP) gives the PP that a variant is causal for a subset of the groups (Figure 1a, Methods; Supplementary Methods).”

And we added the following to the Methods section:

“In flashfm, we obtain marginal model PPs for each trait by fixing each trait and summing over the joint trait PPs for that trait. For example, when there are two traits, the PP for model M_i of trait 1

is the sum over all joint PP for each of the models for trait 2 with model M_i for trait 1. For MGflashfm, after obtaining the multi-group multi-trait joint model $PP_{M_1, M_2}^{(1,2)}$, as above, we calculate the multi-group model PP (mgPP) for each trait that gives the PP for a set of variants C , consisting of variants in a multi-group model. This encompasses all group 1- group 2 models that share at least one variant and C is the collection of all variants in these models. So, the mgPP for a set C is the sum over all combinations of group 1 and group 2 models that share at least one variant and the union of variants is C (Supplementary Methods):

$$mgPP_C = \sum_{i,j \in S} PP_{ij}^{(1,2)}; S = \{(i,j): M_i^{(1)} \cup M_j^{(2)} = C, M_i^{(1)} \cap M_j^{(2)} \neq \emptyset\}.$$

Finally, for variant s , multi-group MPPs are found from

$$MPP_s = \sum_{C: s \in C} PP_C."$$

Minor comments:

1. Can the author provide some intuition if the BF works under cases $MF < 0.05$ and in which MAF the BF approximation is not correct?

Thanks, we have added in some recommendations for MAF thresholds when using these methods, which have some dependence on sample size.

We have added the following to the Methods:

“As these BF estimates depend on the GWAS summary statistics within each group, they will be correct at the same MAF thresholds for inclusion of variants in the GWAS. In general, for larger samples ($\geq 10,000$) we recommend using this approximation for variants with $MAF \geq 0.001$, and for smaller samples to use $MAF \geq 0.005$.”

2. I was under the impression that BF approximation works under the one causal variant but authors have shown that their method works under 2 causal variants. Can they provide some intuition how the BF approximation works under more than one causal variant?

Thank you for this comment which helps to clarify the MGflashfm framework. The Wakefield BF is a common approximation that works for only one causal variant, but we have used a different BF approximation that uses the BIC. This allows BF approximation for multi-SNP models. We have added more detail (as available in the flashfm Supplementary Information), including the explicit form of our BF approximation, to the MGflashfm Supplementary Information (as detailed in our response to comment 4, above).

Reviewer #2 (Remarks to the Author):

In the manuscript "Leveraging information between multiple population groups and traits improves fine-mapping resolution" Zhou et al. improve fine-mapping of variants by jointly analyzing all population groups. The authors put a lot of effort in investigating the performance of the proposed method(s). We commend them for this job while raising one major and two moderately important points.

Thank you for these encouraging comments and we appreciate your helpful suggestions that have improved this paper.

Major points:

1) The authors seem concerned largely only with limiting false negatives (coverage statistics). However, for the best calibration possible there is a need to also concern ourselves with false positives. Consequently, the authors should also provide the comparison of false positive rates, e.g., what fraction of # of causal SNP do the false signals that are selected by methods.

We have added in power and FDR results between all methods (MGflashfm, MGfm, msCAVIAR, PAINTOR, flashfm-EUR, flashfm-AFR) for the small region, and also include comparisons between MGflashfm, MGfm, mvSUSIE-EUR, mvSUSIE-AFR, flashfm-EUR, flashfm-AFR, using two different thresholds for mvSUSIE. We found that MGflashfm, MGfm, flashfm-EUR, flashfm-AFR have similar FDR, whilst PAINTOR has a substantially larger FDR, near 0.75; msCAVIAR has FDR=0, but it also has power=0 because all variants have low PP with mean PP near 0.3 for causal variants. mvSUSIE applied to AFR, has similar FDR to MGflashfm, MGfm, flashfm-EUR, flashfm-AFR, and mvSUSIE applied to EUR results in slightly higher FDR.

We have added the following to Results, as well as a new figure (Fig 4), and two new Supp Figures 2 and 3:

“Next, we examined the false discovery rate (FDR), defined as the mean proportion of non-causal variants having PP above a certain threshold (e.g. 0.5, 0.9), and the power, defined as the mean proportion of causal variants having PP above a certain threshold (e.g. 0.5, 0.9). For 2-trait simulations, where traits each have two causal variants, of which one is shared (T1: AD, T2: AC), we compared power and FDR within two (EUR-AFR, EUR-EAS) and three (EUR-EAS-AFR) ancestries with unequal sample sizes (90,000-10,000; 90,000-40,000-10,000).

There is a general pattern of highest power for MGflashfm and MGfm, and similarly high powers for the group-specific flashfm and PAINTOR (Supplementary Figure 2). The power and FDR for msCAVIAR are consistently 0, so not included in the plots. This is due to msCAVIAR’s uniformly low PP for causal variants, with the mean PP for causal variants near 0.3. The FDR of MGflashfm, MGfm, and the group-specific flashfm are similarly low at PP threshold 0.9, but PAINTOR has very high FDR, of similar magnitude to its power (Supplementary Figure 3).”

And also:

“Next, we include comparisons with a new multi-trait fine-mapping method, mvSUSIE¹⁷, for each population group to confirm the advantages of multi-group multi-trait fine-mapping with MGflashfm. We simulate two traits (each with two causal variants, of which one is shared, i.e. T1: AD, T2: AC) for EUR-AFR with sample sizes of 90,000 and 10,000, using the region of 1610 variants and compare power and FDR for MGflashfm, MGfm, flashfm-EUR, flashfm-AFR, mvSUSIE-EUR, and mvSUSIE-AFR. We note that mvSUSIE returns cross-trait PIP (posterior inclusion probability) and does not return trait-specific PIP that would be analogous to the MPP of the flashfm methods. To infer which variants affect particular traits, mvSUSIE outputs the lfsr (local false sign rate) for each variant under each trait; like p-values, small values indicate an

impact on the trait and we use lfr thresholds of 0.01 and 0.1, as there is no clear mapping between the two types of thresholds. We use MPP threshold 0.9 and find that MGflashfm and MGfm have the highest power, whilst the multi-trait methods flashfm and mvSUSIE have similar power (Figure 4). The FDR are generally similarly low for all methods, but slightly higher for mvSUSIE-EUR and lowest for flashfmAFR (Figure 4); there are longer LD blocks within EUR than in AFR.”

Moderate points:

1) Figure 1 is hard to read. You should find a way to get the text inside the text boxes to be larger

Thank you for this comment. We have edited Figure 1 to improve readability.

2) Edit the paper to improve readability further, e.g., eliminate some duplicated words within sentences/adjacent sentences.

Thank you for this comment and we have carefully gone through the paper to make it more concise - all edits are tracked.

REVIEWERS' COMMENTS

Reviewer #1 (Remarks to the Author):

The authors have answered all my concerns.

Reviewer #2 (Remarks to the Author):

The authors addressed my comments.